# Myotonic Dystrophies: A Genetic Overview

**DOI:** 10.3390/genes13020367

**Published:** 2022-02-17

**Authors:** Payam Soltanzadeh

**Affiliations:** Department of Neurology, David Geffen School of Medicine at UCLA, Los Angeles, CA 90095, USA; psoltanzadeh@mednet.ucla.edu

**Keywords:** myotonic dystrophy type 1 (DM1), myotonic dystrophy type 2 (DM2), muscular dystrophies, nucleotide repeat expansion disorder, spliceopathy, dystrophia myotonica protein kinase (DMPK), cellular nucleic acid-binding protein (CNBP), zinc finger protein 9 (ZNF9)

## Abstract

Myotonic dystrophies (DM) are the most common muscular dystrophies in adults, which can affect other non-skeletal muscle organs such as the heart, brain and gastrointestinal system. There are two genetically distinct types of myotonic dystrophy: myotonic dystrophy type 1 (DM1) and myotonic dystrophy type 2 (DM2), both dominantly inherited with significant overlap in clinical manifestations. DM1 results from CTG repeat expansions in the 3′-untranslated region (3′UTR) of the *DMPK* (dystrophia myotonica protein kinase) gene on chromosome 19, while DM2 is caused by CCTG repeat expansions in intron 1 of the *CNBP* (cellular nucleic acid-binding protein) gene on chromosome 3. Recent advances in genetics and molecular biology, especially in the field of RNA biology, have allowed better understanding of the potential pathomechanisms involved in DM. In this review article, core clinical features and genetics of DM are presented followed by a discussion on the current postulated pathomechanisms and therapeutic approaches used in DM, including the ones currently in human clinical trial phase.

## 1. Introduction

Myotonic dystrophies (DM) are the most common muscular dystrophies in adults, with an estimated prevalence of 1/8000 [1]. There are currently two genetically defined types of DM—type 1 (DM1) and type 2 (DM2)—both of which are autosomal dominant systemic diseases that can involve multiple organs. DM1 and DM2 are considered muscular dystrophies and are usually managed by neuromuscular specialists; however, their non-muscular complications, such as cardiac, gastrointestinal and sleep abnormalities, require a multidisciplinary team approach [2]. DM1 and DM2 are both nucleotide repeat expansion disorders and are thought to share the pathomechanisms involved in other neurodegenerative diseases, such as some forms of spinocerebellar ataxias and amyotrophic lateral sclerosis (ALS) [3]. DM1 is most common in individuals of European descent, but is also present in Japan, China, India and the Middle East [4]. The majority of DM2 patients are of Northern European/German descent and studies suggest a single founding mutation in DM2 patients of European origin dating back to ~200–540 generations [5]. DM1 and DM2 share many clinical features but are caused by two different genes. DM1 results from CTG repeat expansions in the 3′-untranslated region (3′UTR) of the *DMPK* (dystrophia myotonica protein kinase) gene on chromosome 19 [6,7] while DM2 is caused by CCTG repeat expansions in intron 1 of the *CNBP* (cellular nucleic acid-binding protein), previously known as *ZNF9* (zinc finger protein 9), on chromosome 3 [8,9]. In this article, we will have an overview of core clinical features in DM, the genetics of DM1 and DM2, postulated pathomechanisms in DM, and potential therapeutic approaches.

## 2. Clinical Features

Clinical presentations of myotonic dystrophies have a wide spectrum, from asymptomatic or pauci-symptomatic DM2 patients to severe neuromuscular and cognitive deficits in congenital DM1. Myotonic dystrophies are systemic diseases and their clinical manifestations are not limited to skeletal or cardiac muscles (Table 1). Sometimes, the non-muscular features precede myopathic features and can be a much more prominent source of disability compared to the muscular features. Such non-muscular features include central nervous system involvement leading to sleep, cognitive and behavioral disorders, as well as gastrointestinal complications. In DM1, higher numbers of CTG repeats are usually associated with an earlier onset of disease and more severe phenotype [10,11,12,13]. In contrast, higher numbers of CCTG repeats in DM2 do not correlate with a more severe phenotype [14,15]. In DM2, there is no congenital form and the presentation can be more variable or subtle, making the diagnosis more challenging. Anticipation is also noted only in DM1, sometimes leading to the most severe form of the disease called congenital DM1. Children with congenital DM1 are usually born to mothers who themselves may have a mild form of classic DM1 [11,16]. Anticipation is believed to occur because CTG repeats greater than 34 are unstable and can form secondary structures that are not repaired during meiosis, leading to longer repeats over successive generations [17]. In this section of the paper, clinical features of myotonic dystrophies are briefly discussed.

**Table 1 genes-13-00367-t001:** Clinical manifestations of myotonic dystrophies type 1 (DM1), congenital DM1, and myotonic dystrophy type 2 (DM2) [14,15,18,19].

Clinical Feature *	Congenital DM1 (CDM1)	DM1	DM2
Cognitive dysfunction	Usually severe	Can be prominent	Not apparent
Behavioral disorder	Prominent	Can be prominent	Not apparent
Sleepiness	Severe	Prominent	Not apparent
Early cataract	Not seen in childhood	Almost always present	Can be present
Facial weakness	Usually severe	Prominent	Rare
Bulbar weakness	Prominent	Prominent	Rare
Proximal weakness	Present	Usually absent or mild	Prominent
Distal weakness	Usually prominent	Can be prominent	Uncommon
Myotonia on EMG	Usually absent in childhood; develops later	Almost always present	Variable; can be absent
Muscle pain (myalgia)	Usually absent	Usually absent	Present in many
Respiratory failure	Can be severe	Can be prominent	Rare
Tremors	Absent	Absent	Present in many
Cardiac dysrhythmias	Can be absent in childhood; develops later	Almost always present	Variable, can be severe
Gastrointestinal symptoms	Can be severe	Can be prominent	Rare
Diabetes mellitus	Can develop later	Frequent	Variable, can be prominent
Other endocrinopathies	Can be absent in childhood	Can be prominent	Variable, can be prominent
Life expectancy	Reduced	Reduced	Normal

* These clinical features have been oversimplified for comparison purposes. There is phenotypic overlap between DM1 and DM2 and each category has a wide clinical spectrum.

### 2.1. Myotonic Dystrophy Type 1 (DM1)

Clinical features of classic DM1 (Steinert’s disease) include muscle weakness in the distal extremities, leading to the inability to perform fine motor tasks with the hands, and foot drop, leading to falls. In more severe cases, proximal muscles may be involved, causing difficulty with climbing stairs and reaching up above the head [11,19]. Weakness in the cranio-bulbar segment causes a change in voice (dysarthria), dysphagia, obstructive sleep apnea, facial weakness and ptosis. Myotonia in the hands can impair dexterity. Myotonia can also involve cranial muscles, such as the tongue and chewing muscles, and impair their function and cause discomfort. Neuromuscular respiratory insufficiency combined with aspiration pneumonia can at times be a presenting feature in DM1 and lead to prolonged intensive care unit stays or the need for invasive ventilation [20].

Cardiac complications are a major cause of unexpected morbidity and mortality and can include various dysrhythmias and heart blocks. DM1 patients need regular cardiac surveillance and if needed, pacemaker placement or other interventions. Cardiac pump malfunction can also be seen in DM1 [21].

Central nervous system involvement leads to cognitive/intellectual and/or behavioral impairment as well as central sleep disorders [22]. Eye involvement presents with early cataracts. Endocrinopathies in the form of diabetes, thyroid dysfunction and hypogonadism can be prominent. Gastrointestinal symptoms include dysphagia, constipation/pseudo-obstruction and gall bladder stones. Fatigue is a very prominent feature in DM1 and is multifactorial [2,23]. Chronic poor quality or inadequate sleep, myotonia, muscle pain or other chronic pains, as well as central nervous system disorders can all contribute to the development of fatigue.

### 2.2. Congenital DM1

Patients with congenital DM1 show the most severe form of DM. Congenital DM1 presents with reduced fetal movements, polyhydramnios, congenital hypotonia and contractures, difficulty with sucking or swallowing during the first month of life and congenital/infantile neuromuscular respiratory insufficiency [2,11,23]. Surviving infants typically show a gradual improvement in motor function. There are delays in cognitive and motor milestones and all such children have learning disabilities, requiring special education. Clinical or electromyographic myotonia is not a prominent feature in congenital DM1 during childhood. These children can develop other features seen in classic DM1 later in life, including myotonia and cardiorespiratory complications [2,23].

Childhood-onset DM1 is often defined with an onset of symptoms from 1 to 10 years of age and with an intermediate phenotype between congenital DM1 and classic DM1. These children can have gastrointestinal symptoms, learning disabilities, facial weakness, myotonia and cardiac conduction defects [2,23].

### 2.3. Myotonic Dystrophy Type 2 (DM2)

DM2, which was originally called proximal myotonic myopathy (PROMM) has significant clinical overlap with DM1 but tends to have a milder phenotype and can be harder to recognize compared to DM1 [11]. The main clinical distinction from DM1 is the lack of congenital and childhood onset forms. There are no distinct subgroups in DM2, and the symptoms can be severe in early adult-onset form or very mild in late-onset form; however, in some patients, clinical features such as sleep disorders can start very early in life [14,18]. Presenting symptoms include early-onset (younger than 50 years old) cataracts, grip myotonia, proximal leg weakness, thigh stiffness and pain, and frequent falls due to poor balance. Hip flexor, hip extensor and deep finger flexor muscle groups are more likely to be involved in DM2 [14,18]. Facial weakness and ankle dorsiflexor weakness (foot drop) are not usually observed and significant intellectual/learning disability is typically absent [14]. Muscle pain (myalgia) can be more prominent in DM2 than in DM1 and many patients may originally be diagnosed with fibromyalgia. Study of a large cohort of DM1 and DM2 patients showed that the frequency and severity of cardiac involvement and muscle weakness (based on manual muscle testing) are less in DM2 compared to DM1, and that progression is slower; however, careful cardiac evaluation to identify patients at risk for potential major cardiac arrhythmias is recommended [24]. Mild late-onset forms of DM2 are more difficult to diagnose since lower extremity deficits are attributed to degenerative spine disease, diabetic neuropathy and/or age-related deconditioning. In some patients, clinical myotonia or electromyographic myotonic discharges are absent or only observed in a few muscles [15]. Lack of prominent myotonia on EMG and presence of other comorbidities make diagnosis of DM2 difficult. In patients who present with prominent proximal limb weakness, the clinician may only consider limb-girdle muscular dystrophies (LGMD) and order a sequencing panel of LGMD genes, which would not identify the genetic culprit. An epidemiological study from Finland and Italy suggests that DM2 is more frequent than DM1 and challenges in identification of minimally affected individuals may lead to underestimation of DM2 prevalence [1].

## 3. Genetics of Myotonic Dystrophies

DM1 and DM2 share many clinical features but are caused by two different genes. DM1 results from CTG repeat expansions in the 3′-untranslated region (3′UTR) of the *DMPK* (dystrophia myotonica protein kinase) gene on chromosome 19 [6,7], while DM2 is caused by CCTG repeat expansions in intron 1 of the *CNBP* (cellular nucleic acid-binding protein), previously known as *ZNF9* (zinc finger protein 9), on chromosome 3 [8,9] (Table 2). In DM1, there is a correlation between the number of CTG repeats and the severity and age of onset of the disease [10,11,12,13]. Congenital DM is the most severe form of DM1 and is usually seen in infants with CTG repeat size range of 750 to 1400 or higher [11,23,25]. Milder phenotypes such as classic DM1 and the more severe childhood-onset DM1 show a lower number of CTG repeats typically starting from 50 and above (Table 3). Individuals with CTG repeat numbers between 38 and 50 are considered to have a “pre-mutation” allele and may remain asymptomatic throughout their lives. Children of individuals with “pre-mutation” are at increased risk of having larger CTG repeats and potential to develop symptoms. This stems from the bias towards CTG repeat expansion (instability) in successive generations leading to “anticipation”, where children of DM1 patients have a larger repeat number of CTG repeats, earlier onset of disease, and a more severe phenotype [16]. Based on peripheral blood leukocyte analyses, CTG repeat “contraction” from parent-to-child transmission can also occur in about 6.4% of DM1 patients, with most contractions occurring in paternal transmissions [26]. In 3 to 5% of DM1 families, there are GGC, CCG and CTC interruptions within the CTG repeat array potentially leading to repeat stabilization and milder phenotypes. Although anticipation occurs in DM1 families with these variant repeats (VRs), congenital DM1 has not been reported [27,28]. VRs have been associated with reduced somatic instability and a milder DM1 phenotype [13,29,30]. In a recent study of DM1 families from Spain, three sisters from the same family with CCG interruptions, showed atypical clinical traits such as late onset of symptoms (>50 years), severe axial muscle involvement, and lack of typical craniofacial features of DM1 [28]. Such atypical presentations make clinical diagnosis of late onset DM1 difficult and show that the implications of these VRs in the pathogenesis of DM and its phenotypic subtypes require further investigation [13]. Since patients with VRs seem to have a later age of onset and possibly a slower progression of disease, future clinical trials should take them into consideration [13,29]. Altered pattern of methylation around repeat expansions has been described in DM1 patients with VRs, which could potentially affect local gene expression and influence repeat instability [30,31]. The emergence of interruptions may be caused by multiple processes including spontaneous DNA damage, DNA repair and DNA polymerase errors occurring in germ cells and somatic cells throughout embryogenesis and the lifetime of DM1 patients [13,32,33]. The location of these interruptions changed dramatically between generations and the repeats tended to contract [27,31,34]. These stabilizing variants can cause false negative test results in both repeat primed PCR and standard PCR based approaches; therefore, bidirectional triplet primed PCR (TP-PCR) should be used for DM1 testing [33,35,36,37]. The development of new technologies, such as long-read sequencing, will allow, in the near future, better characterization of repeat expansions both at DNA and RNA levels [33,38].

The CTG repeat typically continues to expand during the lifespan of DM1 patients, causing the development of new symptoms or an increase in the severity of symptoms as the individual ages [12,39]. The rate of somatic instability varies in different tissues from the same individual and tissues that are most involved in DM1, such as muscle, heart and brain, demonstrate the largest repeat expansions over time [40]. Ongoing somatic expansions and tissue variabilities may partly explain some discrepancies noted between the severity of DM1 phenotype and measured CTG repeat numbers. CTG repeat measurements are based on DNA extraction from peripheral blood, a tissue with much lower somatic instability compared to more vulnerable tissues such as skeletal muscle, heart, or brain. For example, the repeat length is always much larger in muscle DNA than that observed in blood [41]. To improve the correlation between CTG repeat length measurements and age at onset, and to correct for age-at-sampling biases, methods have been proposed to estimate the progenitor cells CTG length and somatic mutational dynamics using small pool PCR. An accurate measurement of the estimated progenitor allele length (ePAL) is currently the best predictor of age at symptom onset [42,43]. Mismatch repair machinery (MMR) has been shown to be responsible for fixing replication errors made during replication slippage within microsatellites [44]. Polymorphisms in the MMR gene *MSH3* have been identified as a modifier of somatic instability in DM1 patients, suggesting this gene as a potential molecular target in repeat expansion diseases [45,46].

Congenital DM1 (CDM1) is an extreme example of the parent-of-origin effect since most CDM1 children receive the CTG repeat expansion from their mothers [47]. CTG lengths between 50 and 500 show a paternal expansion bias, while for CTG >500 repeats, maternal transmission shows the stronger expansion bias to exceedingly large repeat expansions (>1000–6500 repeats), often leading to CDM1 [25]. The majority of DM1 patients who inherit expansions from their father do not develop CDM1, even if the repeat length is greater than expansion lengths typically found in CDM1, suggesting that other factors, in addition to the CTG repeat length, contribute to the most severe symptoms and to the maternal transmission bias observed in CDM1 patients [25]. Many mothers of CDM1 children, are pauci- or asymptomatic and diagnosed with DM1 only after giving birth to their affected babies. To explain the parent-of-origin effect in CDM1, it is believed that sperms with large CTG expansions possibly die prematurely, are somehow incapable of fertilizing the oocytes or the resulting embryos are not viable. Methylation/demethylation dysregulation has been suggested as a possible explanation for the parent-of-origin effect in CDM1. Methylation and demethylation events can be differentially regulated in oogonia and spermatogonia and the presence of CTG repeat expansions may disrupt such regulatory mechanisms. This has been shown in fragile X syndrome, in which full mutations arise exclusively on transmission from a mother who carries a premutation allele (60–200 CGGs) [48]. CpG hypermethylation of a region upstream of the CTG expansion in cells of CDM1 tissues has been associated with CDM1 [49]. This region of hypermethylation contains a binding site for the insulator transcription factor CTCF (CTCF1 site). Two binding sites for the insulator protein, CTCF, surround the CTG repeat and regulate anti-sense transcription as well as replication fork progression. CpG methylation of CTCF1 prevents binding of CTCF, potentially affecting chromatin dynamics [25,49]. In the study by Barbe et al., CpG methylation flanking the CTG repeat was analyzed in 79 blood DNAs from DM1 individuals, of which 20 had CDM1. Almost all (19/20) patients with CDM1 had DNA hypermethylation at the CTCF1 binding site, while only 4% of classic DM1 individuals and none of the unaffected controls had DNA methylation at the same site [49]. Another hypothesis to explain the maternal bias in CDM1 is the effect of CTCF1 site hypermethylation on the expression of the homeobox gene *SIX5*, located immediately downstream of the *DMPK* gene, in the male germ line. Hypermethylation of CTCF1 site can lead to reduced expression of Six5 protein. Six5 protein is involved in spermatogenesis survival and spermiogenesis and its suppression may lead to reduced survival of male germ line cells with CDM1 progeny [25].

In DM2, the smallest reported mutations in CNBP vary from 55 to 75 CCTG repeats and the largest are about 11,000. Similar to DM1, there is significant somatic instability with increase of the CCTG repeat length over the lifespan of the individual [18]. DM2 repeat tract contains the complex repeat motif (TG)n(TCTG)n(CCTG)n. The CCTG portion of the repeat is interrupted on normal alleles, but these interruptions are lost on affected alleles leading to further expansions [50]. As opposed to DM1, in DM2, the mutation usually contracts in the next generation, with the affected offspring having markedly shorter expansions compared to their affected parents, and repeat size did not correlate with age at disease onset [14]. In DM2, there is not a congenital or childhood-onset form even with very large numbers of CCTG repeats. This indicates that other factors besides repeat numbers are playing a role in myotonic dystrophies. The size of the CCTG repeat expansion from peripheral blood white cells in DM2 seems to correlate with the age of the patient but not necessarily with the severity of the phenotype [11]. Extreme somatic instability has been illustrated in monozygotic twins [14].

Similar to DM1, a bidirectionally labelled PCR method is recommended to avoid false negative results caused by CCG, CTG, and GGC sequence interruptions; however, due to large CCTG repeat size range in DM2 (75 to >11,000) and somatic instability, genomic southern analyses fail to detect expansions in 20% of known carriers. Quadruplet-repeat primed (QP)-PCR has been used with an additional primer within the elongated CCTG repeat to detect expansions independent of their lengths. Although expansions in all size ranges can in theory be detected by QP-PCR, no reliable information about the length of the expanded repeat will be obtained because of extinction of the signal in the higher size region [14,37]. Due to variability of the complex (TG)n(TCTG)n(CCTG)n repeats, the precise length of the pathogenic CCTG unit within this repeat can only be determined by sequencing. Since the difference between normal alleles (repeats with 26–30 CCTG units) and disease-associated alleles (75 units and more, with a mean of 5000) is almost always evident, exact sizing is not routinely performed. Alleles with 26 to 75 CCTG units (“gray area”) are very rare, but their clinical relevance is yet to be determined [14,36,37,51].

## 4. Postulated Pathomechanisms

Expansions of DNA repeat sequences cause numerous neurological and neuromuscular diseases including Huntington’s disease, fragile X syndrome type A (FRAXA), Friedreich’s ataxia (FRDA), spinocerebellar ataxias (SCAs) and myotonic dystrophies (DM). In unaffected people, these DNA repeats are short and stable, showing no significant length changes; however, in affected individuals, the tracts are longer and unstable. These length changes are dynamic across generations, over the life span of an individual, and also between and within tissues from the same individual. Formation of unusual DNA, DNA-RNA, and RNA structures and their interference with nucleic acid replication, repair, recombination and transcription have been suggested as processes whereby DNA is damaged, repeats expand or contract, mis-splicing occurs, proteins are sequestered and, ultimately, translation is disrupted [12,39]. Transcription and repair models of repeat instability can account for expansion in both mitotic and post-mitotic cells; however, since disease manifestations are more pronounced in postmitotic tissues (brain, heart, and skeletal muscle), DNA repair-dependent mechanisms and potentially transcription-coupled nucleotide excision repair seem to be more important in driving repeat instability in these tissues of patients with DM [12].

Different mechanisms have been suggested to explain the various pathologies in DM1 and DM2 (Table 4). There is much more literature on potential implications of CTG repeat expansions in DM1 than of CCTG expansions in DM2; however, there could be common mechanistic pathways since both diseases are caused by short nucleotide repeat expansions and have many clinical features in common. Experimental studies support an RNA gain-of-function mechanism (RNA toxicity hypothesis), whereby CUG/CCUG-containing RNAs form ribonuclear foci that sequester and disrupt nuclear factors required for proper muscle development and maintenance, including those involved in regulation of alternative splicing.

### 4.1. Alternative Splicing Defects

The most prominent pathomechanisms in DM1 and DM2 are thought to be abnormalities in alternative splicing. Alternative splicing (AS) is a regulatory mechanism of gene expression allowing the production of more than one unique mRNA from a single gene. Splicing regulation in skeletal muscle is crucial for normal myogenesis and adaptation to changes in post-natal metabolic and functional requirements. Splicing occurs in spliceosomes, which are large nuclear complexes made of smaller nuclear ribonucleoprotein particles and many regulatory factors including RNA-binding proteins (RBPs). RBPs are involved in inclusion or exclusion of different exons in each transcript depending on the specific cell context, cell type and tissue-specific developmental processes. Muscle-blind-like (MBNL) family, CUG-BP and ETR-3-like factors (CELF) family and the RNA-binding Fox (RBFOX) are the most important splicing regulators in skeletal muscle [52,53]. MBNL proteins seem to play an important role in DM1 pathogenesis and their activity is controlled by RNA secondary structures [54]. MBNL1, which is the most abundant MBNL in adult skeletal muscle, plays an important role in splicing regulation in the cardiac and skeletal muscle tissues. It is believed that more MBNL units are trapped with a larger CUG expansion, hence more potential consequent toxicity. Retention of both MBNL and DMPK CUG repeats in the nucleus decreases functional concentrations of DMPK and MBNL in cytoplasm and over time leads to impaired translation. *MBNL1* knockout mice show similar features characteristic of DM1 disease [55]. MBNL2 seems to be more crucial for splicing regulation in the brain tissue and *MBNL2* knockout mice exhibit a number of DM-related central nervous system abnormalities [56]. In addition to their role in alternative splicing, MBNLs also seem to be involved in regulation of mRNA localization and stability [57].

CELF1 (previously called CUGBP1), a member of the family of CELF (CUGBP, Elav-like family) proteins, is another RBP that seems to play a role in DM1 pathogenesis. CELF1 binds to single-stranded UG-rich RNA sequences but not to double-stranded sequences, unlike MBNL1, suggesting a key difference in their splicing regulatory mechanisms. CELF1 binds to CUG-repeat containing RNA with a lower affinity than MBNL1 and loss of functional MBNL1 induces CELF1 upregulation. CELF1 overexpression in MBNL1-depleted DM1 tissues leads to expression of embryonic variants of transcripts, the main characteristic of spliceopathy in DM1 [58,59]. MBNL1 and CELF1 act as antagonist regulators of several pre-mRNA targets, including cardiac troponin (cTNT), insulin receptor (INSR), chloride channel 1 (CLCN1) and MBNL2. Transcript alterations in various tissues are detailed in the review by Lopez-Martinez et al. [53].

Intranuclear accumulation of splicing and cleavage factors, such as heterogeneous nuclear ribonucleoproteins (hnRNPs) and small nuclear ribonucleoproteins (snRNPs) has been shown in both DM1 and DM2 patient biopsies [60]. These factors are needed for the pre-mRNA to generate a mature transcript in the spliceosome [53].

Dysregulation of stress granule (SG) dynamics has been suggested as an important pathogenic factor in certain neurodegenerative diseases. SGs are transient cytoplasmic cell compartments that are formed in response to different stress stimuli and are implicated in the regulation of translation, mRNA storage and stabilization, and cell signaling during stress. They are formed by coalescence and storage of translation factors, mRNAs, RNA-binding proteins (RBPs) and other proteins that facilitate cell survival during stress. Upon resolution of stress, SGs should disperse; however, in age-related and neurodegenerative disorders such as amyotrophic lateral sclerosis (ALS), rigid complexes are formed in SGs potentially leading to accumulation of irreversible toxic aggregates [61]. Altered response to stress and abnormal clearance of SGs caused by MBNL1/CELF1 disruption have been suggested as one of the pathomechanisms in DM1 and treatments that selectively target pathological SGs might be considered a potential therapeutic strategy [62,63].

### 4.2. Repeat-Associated Non-ATG (RAN) Translation

RAN translation has been described in several different types of repeat expansion mutations (CAG, CTG, CCG, GGGGCC, GGCCCC) and results in the production and accumulation of potentially harmful proteins. It has been shown that expansion mutations can be bidirectionally transcribed [64]. Repeat expansion disorders, including DM1 and DM2, have been shown to be bidirectionally transcribed [65,66]. RAN proteins are found in a variety of CNS cell types. Three mechanisms of loss-of-function, RNA gain-of-function, and protein gain-of-function have been proposed. In DM1, it is thought that translation of multiple reading frames occurs and produces various protein aggregates in both the nucleus and cytoplasm. One of such proteins is polyglutamine, which results from DM1 antisense transcription (CAG repeats) [65,67]. Accumulation of these aggregates can lead to apoptosis. Larger repeat sizes are associated with more efficient RAN translation, and possibly more severe phenotype [65,67]. Abnormal accumulation of tetrapeptide poly-(LPAC) and poly-(QAGR) RAN proteins have been shown in DM2 autopsy brains. LPAC and QAGR proteins are toxic to cells [66]. The role of RAN translation in DM pathogenesis needs further investigation.

### 4.3. microRNA Dysregulation

microRNAs (miRNAs) are small non-coding RNAs that modulate gene expression at the post-transcriptional level and their expression and intracellular distribution is dysregulated in many human diseases, including muscular dystrophies. miRNAs promote messenger RNA (mRNA) degradation, destabilization or translation blocking with participation of additional protein factors, forming the RNA-induced silencing complex (RISC). It is estimated that about 2000 miRNAs identified in humans can modulate up to 60% of protein-coding genes [68]. Some miRNAs biomarkers have been identified in Drosophila and mouse models of DM1, which seem to be translational repressors of MBNL1/2 and MBNL2 [69,70,71]. A group of miRNAs have been detected in the peripheral blood of DM1 and DM2 patients that may play a role in DM pathogenesis and can potentially serve as serum biomarkers. A “DM1-miRNA score” has been used to identify DM1 patients from a control group [68,72,73,74]. Serum miRNA biomarkers have the advantage of being derived from various tissues and not solely from the skeletal muscle, hence potentially reflecting the global clinical state of the patient. MBNL1 has been shown to be a regulator of pre-miR-1 biogenesis and dysregulation of MBNL1 and miR-1 interactions has been demonstrated in the heart tissue of patients with DM1 and DM2, suggesting a possible role of this miRNA in cardiac involvement in DM [75]. Another study of six selected miRNA candidates did not show consistent differences between sera of DM1 patients versus sera from controls [76]. Discrepant results were thought to be due to different techniques and sample qualities. Clinical implications of circulating miRNAs in DM need more studies with larger number of subjects, more diverse control groups, including patients with other types of muscular dystrophies, and standardized experimental procedures. Despite the genetic and clinical similarities, candidate miRNA biomarkers identified so far are different in DM2 compared to DM1 [74,75]. A comprehensive overview of miRNA alterations in DM1 and DM2 is discussed by Lopez Castel et al. [68].

### 4.4. Circular RNAs (circRNAs) Dysregulation

Circular RNAs (circRNAs) are a large class of endogenous non-coding and very stable RNAs produced by a non-canonical splicing event called backsplicing. They have tissue-specific and cell-specific expression patterns and their biogenesis is regulated by specific cis-acting elements and trans-acting factors [77]. It is not known how the splicing machinery decides whether to generate a linear mRNA or a circRNA, and many factors such as a variety of RNA-binding proteins including splicing factors must be involved in their co-transcriptional and post-transcriptional biogenesis [78]. circRNAs are thought to have multiple functions that include serving as microRNA (miRNA) sponges or decoys, protecting target mRNAs from miRNA-dependent degradation, acting as protein sponges, being scaffolds to mediate complex formation between specific enzymes and substrates and recruiting proteins to specific locations. Although circRNAs are typically non-coding, a subset of circRNAs with internal ribosome entry site (IRES) elements and AUG sites may be translated under certain circumstances, giving rise to unique peptides [77]. The presence of circRNAs is more prominent during the differentiation process and circRNAs are highly expressed in mammalian brain, striated muscle, and cardiac tissues. It has been shown that the second exon of the splicing factor muscleblind (MBL/MBNL1) is circularized in flies and humans forming circMbl. This circRNA and its flanking introns contain conserved muscleblind binding sites, which are strongly and specifically bound by MBL/MBNL1. It is thought that circularization and splicing compete against each other and an autoregulatory circuit may exist in which excess MBL or MBNL1 decreases the production of its own mRNA by promoting circRNA biogenesis, and the circRNA promotes the linear splicing of the gene by tethering MBL or MBNL1 [78,79]. Other regulatory proteins have also been implicated in the biogenesis and regulation of circRNAs via various mechanisms. These include Quaking (QKI), Fused in Sarcoma (FUS), hnRNPs (heterogeneous nuclear ribonucleoproteins) such as Hrb87F, SR (serine-arginine), andADAR1 (adenosine deaminase acting on RNA 1) proteins [78]. Study of a selected number of circRNAs in muscle biopsies of patients with DM1 suggested dysregulation of circRNA expression in DM1. In this study, silencing of either MBNL1 or CELF1 in the control and DM1 cultured myotubes did not affect the abundance of circZNF609, circRTN4, and circRTN4_03 and the authors explained this finding by the possible role of other splicing regulatory factors, or higher stability of circRNAs compared to their linear counterparts [80]. In the aforementioned study as well as the study by Czubak et al. [81], upregulated levels of circRNAs were correlated with the severity of the patients’ muscle phenotype. Some of the circRNA changes have also been identified in other muscular dystrophies and the specificity of such abnormalities in the splicing machinery needs to be determined with future studies. circRNAs may contribute to the pathogenesis of some neurodegenerative diseases due to their high stability, their accumulation over time in non-dividing cells, and their interactions with other RNAs and proteins. Since circRNAs are highly resistant to exonucleases and can be retrieved from serum or plasma, they could potentially be useful disease biomarkers.

### 4.5. Epigenetic Modifications

Epigenetic modifications may impact the development or severity of phenotype in DM1. This has been well-characterized in congenital DM1 with the role of DNA hypermethylation at the CTCF1 binding site, which was discussed above in the genetics section of this review article. A recent review article summarizes available evidence on the potential role of epigenetics in DM1 and DM2 [82]. DNA methylation upstream of the CTG repeats in blood DNA has been proposed as a marker of maternally inherited congenital DM1 [25,49]. In a recent study of 90 adult DM1 patients, DNA methylation downstream of the CTG array in the *DMPK* gene was correlated with the phenotypic variability in several muscular and respiratory parameters; this association was not linked to the CTG repeat length [83]. Another study of DM1 subjects showed that in patients without variant repeats (VRs), blood DNA methylation at baseline correlated with cognitive function 9 years later, independent of the number of CTG repeats; and patients with VRs (12 out of the total number of 115) had different DNA methylation and cognitive profiles [84]. Genetic and phenotypic features of reported individuals with VRs in the DMPK (CTG)n array have been summarized in a recent review by Meola’s group [32]. Patients with VRs have a different DNA methylation profile with higher downstream hypermethylation; however, the role of VRs in DM1 phenotype needs to be investigated by further studies.

### 4.6. Cellular Nucleic Acid Binding Protein (CNBP), Formerly Known as Zinc Finger Protein 9 (ZNF9)

Cellular nucleic acid binding protein (CNBP) is a small human single-stranded nucleic acid-binding protein and is involved in transcription regulation by binding to both single-stranded DNA and RNA molecules and by activating or repressing the expression of many genes [85]. A heterozygous *CNBP* knock-out (*Cnbp* −/+) mouse model showed some of the multi-organ phenotypic features of myotonic dystrophy, including muscle histopathology, myotonic discharges and heart conduction abnormalities. The crossing of these mice with transgenic mice overexpressing CNBP increased CNBP and CLCN1 levels and rescued the phenotype, suggesting that, in addition to RNA gain of function, a loss of CNBP may contribute to DM2 pathology [86]. However, using patient-derived myoblast cell lines and DM2 skeletal muscle biopsy tissue, Margolis et al. showed that pre-mRNAs containing large CCUG expansions are normally spliced and exported from the nucleus, that the expansions do not decrease CNBP expression at the mRNA or protein level, and that the ribonuclear inclusions are enriched for the CCUG expansion, but not intronic flanking sequences [87]. This study suggested that, as opposed to DM1, post-transcriptional silencing may not be a reasonable approach to treat DM2 because the downstream effects of the DM2 mutation seem to be different. More DM2-specific experiments are needed to better understand the role of CNBP in DM2 pathogenesis and the downstream effects of CNBP repeat expansions.

### 4.7. CLCN1 and SCN4A Mutations as Possible Modifier Genes in DM2

Mutations in *CLCN1* and *SCN4A* are associated with non-dystrophic myotonic myopathies (skeletal muscle channelopathies); however, studies of these genes in case series or case reports of patients with DM2 from Germany and Italy have suggested their possible role in the development or severity of myotonia and/or muscle pain in some patients with DM2 [88,89,90]. Co-existing mutations in *CLCN1* and *SCN4A* seem to enhance the myotonia and/or muscle pain in DM2 patients.

## 5. Therapeutic Approaches

Based on the evolving knowledge of possible pathogenetic mechanisms involved in DM1 and DM2, various therapeutic approaches may be considered. DM is a slowly progressing disease and the pattern of gene dysregulation, including spliceopathy, could be different in any specific tissue and also vary over the lifetime of the individual. These limitations impair development of accurate molecular outcome measures, which are crucial for initial drug development. Similar to other genetic diseases, limitations of the preclinical animal models and applicability of their biological mechanisms to humans pose a challenge to drug discovery.

A number of therapeutic strategies have been suggested for DM [91] and some of them have been implemented in animal models and more recently in human clinical trials [92,93,94] (Table 5). There are many drugs in the pipelines at various stages of development some addressing the underlying pathophysiology and some trying to improve various complications of DM1 and DM2 [95]. Currently there is no approved disease-modifying therapy for any form of DM. Although DMs are considered muscular dystrophies, an ideal therapy should be able to reach beyond the skeletal and cardiac muscle tissues, including the central nervous system. Central nervous system disorders such as insomnias and cognitive/behavioral abnormalities can be more disabling than muscular or cardiac pathologies [22]. This is more prominent in early-onset forms of DM1 [23].

**Table 5 genes-13-00367-t005:** Therapeutic approaches in myotonic dystrophies.

Therapeutic Approach	Example(s)
Genome editing	CRISPR/Cas9-mediated deletion of CTG expansions [96]
Reducing the transcription of expanded CUG RNA	Pentamidine and related compounds [97]
Post-transcriptional silencing of expanded RNA	Antisense oligonucleotides (ASOs) interventions [92,98,99]; small interfering RNAs (siRNA) techniques [93,100,101,102]; RNA-targeting CRISPR/CAS9 (RCas9) systems [103,104]
Modulation of RNA-binding proteins or their interactions with the expanded RNA	Erythromycin [105]; MBNL1 overexpression mouse models [106,107]
MicroRNA (miRNA) therapies	miRNA interventions in animal models of DM1 using antagomiRs [70,71]
Inhibition of glycogen synthase kinase 3 beta (GSK3beta)	Tideglusib [94,108]
Inhibition of protein kinase C (PKC) to lower CELF1 and its phosphorylation	PKC inhibitors Ro-32-0432 and Ro-31-8220 [109]
Targeting the AMP-activated protein kinase (AMPK)/mammalian target of rapamycin (mTORC1) pathway	AMPK activator 5-aminoimidazole-4-carboxamide ribonucleotide (AICAR); and mTORC1 inhibitor rapamycin [110]

Reducing the transcription of expanded CUG RNA by inhibition of RNA polymerase co-factors or by small molecules that bind to GC-rich repeats is one strategy. Pentamidine and related compounds may work through binding the CTG-CAG repeat DNA to inhibit RNA transcription in cell and mouse models of DM1 [97]. The new gene editing technique of clustered regularly interspaced short palindromic repeats (CRISPR)-associated protein 9 (Cas9) system has created a lot of excitement in the field of genetic diseases, including DM. Provenzano et al. created myogenic cells derived from fibroblasts of DM1 Patients and designed a CRISPR/Cas9 construct to delete CTG expansions. The authors showed removal of repeat expansions, prevention of nuclear foci formation, and improvement in splicing biomarkers in selected clones [96].

Several nucleic acid technologies, primarily using antisense oligonucleotides (ASOs) have been used to achieve post-transcriptional silencing of DMPK [98]. In a transgenic mouse model of DM1, systemic administration of ASOs caused a rapid knockdown of the expanded CUG repeat RNA in skeletal muscle and the effect was sustained for up to 1 year after treatment was discontinued [99]. Based on these results, a phase 1/2a blinded, placebo-controlled study with multiple ascending doses of subcutaneously administered IONIS-DMPK-2.5Rx to 48 adult patients with DM1 was pursued but the amount of target engagement did not achieve the desired therapeutic benefit to treat this disease [92]. Small interfering RNAs (siRNA) targeting CUG repeat tracts have also been used as a post-transcriptional silencing strategy to treat DM1. Despite the existing evidence that RNAi pathways operate primarily in the cytoplasm, lentivirus-delivered short hairpin RNAs (shRNAs) were shown to significantly reduce the nuclear retained mutant DMPK mRNAs in DM1 cells [100]. In a model using skeletal muscle cells isolated from individuals with DM1, engineered human U7 small nuclear RNAs (hU7-snRNAs) targeting the expanded CUG repeats of mutant DMPK transcripts caused degradation of pathogenic DMPK mRNAs [101]. In a transgenic mouse model of DM1, intramuscular injection and electroporation of siRNA resulted in 70–80% downregulation of expanded CUG transcripts, demonstrating nuclear activity of synthetic siRNA in mammalian tissue in vivo [102]. Using an antibody-oligonucleotide conjugate (AOC) platform, a recent phase 1 clinical trial for adult patients with DM1 was carried out. In this trial, a monoclonal antibody that targets transferrin receptor 1 (TfR1) is conjugated with a siRNA (AOC 1001) that is designed to reduce levels of DMPK RNA in skeletal, cardiac and smooth muscle [93]. Other AOC conjugates are also in the preclinical phases to treat muscular dystrophies and other diseases [111]. Batra et al. used an RNA-targeting Cas9 (RCas9) system in an in vitro overexpression model of DM1 to eliminate CUG repeat foci and reversal of alternative splicing defects; the authors also showed efficient elimination of both CUG and CCUG repeat expansion RNAs in myoblasts and fibroblasts prepared from DM1 and DM2 patient biopsies, respectively [103]. Interestingly, they did not observe DNA-level transcriptional disruption or destabilization of the DNA repeats. RNA-specificity of such approaches can obviate the concern about potential permanent off-target genetic lesions with DNA-mediated CRISPR-based therapeutics. The same group later used adult and neonatal mouse models of DM1 and demonstrated that intramuscular or systemic injections of adeno-associated virus (AAV) vectors encoding nuclease-dead Cas9 and a single guide RNA targeting CUG repeats resulted in the expression of the RNA-targeting Cas9 for up to 3 months, redistribution of the MBNL1, elimination of foci of toxic RNA, reversal of splicing biomarkers and amelioration of myotonia [104]. This RNA-targeting Cas9 intervention showed possibility of using such therapeutic strategy in patients with DM or other nucleotide repeat disorders.

In view of the role of MBNL in the pathogenesis of DM, inhibiting interactions between MBNL and toxic RNA, reduction of CELF1 concentration, and dispersion of RNA foci have been used as therapeutic strategies in various in vitro and in vivo models. Nakamori et al. tested 20 FDA-approved small molecules in a DM1 cell model and found erythromycin to have the highest affinity to expanded CUG and a capacity to inhibit its binding to MBNL1. They then used erythromycin in a DM1 mouse model and showed splicing reversal and improvement of myotonia [105]. MBNL overexpression has been used as a strategy to rescue the phenotype but has led to conflicting results in different mouse models of DM1 [106,107].

As discussed above, dysregulation of microRNA (miRNA) has been implicated in DM pathogenesis. miRNA-targeting approaches have been suggested as a treatment strategy in various models of DM1. There are many technical challenges in implementing such treatments in terms of identifying appropriate targets and biomarkers as well as stability and delivery issues. Various approaches for miRNA intervention have been used or suggested: one is to inhibit miRNA by using synthetic antimiR (also called antagomiR) products, such as ASOs that are perfectly complementary to the specific miRNA target; a second approach is to use blockmiRs, which contain a sequence complementary to one of the mRNA (messenger RNA) sequences that serve as a binding site for a microRNA and sterically block the miRNA from binding to the same site, hence preventing degradation or transcription inhibition of the target; the third approach is to use miRNA sponges that contain several tandemly arranged miRNA target sequences (same or different sequences); and the last one is using synthetic double strand miRNA mimics as replacement therapy. These proof-of-concept approaches have been used in in vitro models as well as in drosophila and mouse models of DM1 and further discussed in a review by Lopez Castel [68]. In a mouse model of DM1, subcutaneous administration of an antagomiR upregulated the expression of MBNL1, rescued splicing alterations, and improved grip strength and myotonia [70].

Signaling pathways downstream of CUG/CCUG expanded repeat expansions have also been used in therapeutic experiments. These include targeting glycogen synthase kinase 3 beta (GSK3beta), protein kinase C (PKC) and AMP-activated protein kinase (AMPK)/mammalian target of rapamycin (mTORC1) pathways [109,110,112]. In a DM1 mouse model, GSK3beta levels were shown to increase, and the inhibition of GSK3beta improved the phenotype [112]. In a recent study, a small-molecule inhibitor of GSK3, tideglusib, reduced the mutant DMPK mRNA in myoblasts from patients with adult DM1 and congenital DM1; it also improved postnatal survival and phenotype in a DMSXL mouse model of congenital DM1 [108]. Following some favorable results from a small clinical trial of tideglusib in congenital and childhood-onset DM1, a larger multicenter phase 2/3 study of patients with congenital DM1 is currently under way [94].

## 6. Conclusions

With recent advances in molecular genetic knowledge and techniques, our understanding of genetic diseases such as myotonic dystrophies has significantly improved; however, there is still much more to learn about these diseases and, as more details are revealed, new questions arise about the pathogenesis of DM1 and especially DM2. Disease-modifying therapies such as siRNA are just entering human clinical trials, but efforts to better understand nucleotide repeat diseases and to identify the best therapeutic targets should continue. Short- and long-term efficacy and safety of current therapeutic approaches are yet to be determined and this knowledge gap is even more noticeable for DM2, where animal models and therapeutic trials are lagging behind DM1. Since various potential pathomechanisms are involved in the development or progression of DM1 and DM2, future disease-modifying therapies may need to be used synergistically to target more than one disease mechanism.

## Figures and Tables

**Table 2 genes-13-00367-t002:** Genetics of DM1 versus DM2.

	DM1	DM2
Chromosomal location	19q 13.3	3q 21.3
Inheritance	Autosomal dominant	Autosomal dominant
Culprit gene	*DMPK*	*CNBP* (previously called *ZNF9*)
Repeat motif	CTG	CCTG
Location of the repeat expansion	3′UTR	Intron 1
Normal repeat size	<37	<26–30
Pathologic repeat size and range	>50 to 4000	75 up to 11,000
Phenotypic correlation with repeat size	Yes	No
Anticipation	Yes	No
Severe congenital/childhood form	Present	Absent

**Table 3 genes-13-00367-t003:** Correlation of CTG repeat size and clinical subtype of DM1 [11,12].

Phenotype	Age of Onset	CTG Repeat Length
Congenital DM1	Before/around birth or early infancy	750–1400
Childhood onset DM1	1–10 years	50–1000
Adult onset “classic” DM1	10–30 years	50–1000
Late onset/asymptomatic	20–70 years	50–100
Pre-mutation	Not applicable (asymptomatic)	38–49

**Table 4 genes-13-00367-t004:** Postulated pathomechanisms in DM1 and DM2.

Pathomechanism	DM1	DM2
Formation of nuclear inclusions and sequestration of regulatory proteins	Yes	Yes
Alternative splicing defects	Yes	Yes
Repeat-associated non-ATG (RAN) translation	Yes	Yes
microRNA dysregulation	Yes	Yes
Circular RNAs (circRNAs) dysregulation	Yes	Yes
Epigenetic modifications	Yes (especially in congenital DM1)	No data
Cellular nucleic acid binding protein (CNBP) haploinsufficiency	Not applicable	Possible
*CLCN1* and *SCN4A* mutations as modifying factors	No data	Possible

## Data Availability

Not applicable.

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
