# Peer review of "Myotonic Dystrophies: A Genetic Overview"

_genes, 2022, doi:10.3390/genes13020367_

Round 1

Reviewer 1 Report

In my opinion, it is very appropriate to propose a review work on DM1 and DM2 at this time when different therapeutic strategies, that seem to be quite promising, are emerging and being developed. The work reflects very well the clinical-genetic knowledge that we currently have about these diseases. In addition, the authors appropriately correlate knowledge about the pathophysiological mechanisms of the disease with the concepts underlying therapies in development. In summary, I think this is a good review of DM1 and DM2 that very well collects the most outstanding aspects that are known about these diseases.  However, I would like to suggest some aspects that, in my opinion, will improve the work’s understanding.

  1. Lines 56-58. “Anticipation is also noted only in DM1, leading to severely affected congenital DM1 children born from mothers who may have a mild form of classic DM1”. A short sentence would be included detailing that the anticipation phenomena appears because CTG repeats greater than 34 are unstable and may expand during meiosis at risk of increasing over successive generations.
  2. Line 135: “In DM1, there is a correlation between the number of CTG repeats and the severity of the disease”. Although it is well explained that congenital DM is the most severe form of DM1, I would add in the mentioned sentence that there is a correlation between the number of CTG repeats and the severity and age of onset of the disease.
  3. Line 148: “……interruptions within the CTG repeat array leading to repeat stabilization and potentially milder phenotypes”. It seems that there could be some discrepancies regarding this point. In the following article (Ballester-Lopez A et al. A DM1 family with interruptions associated with atypical symptoms and late onset but not with a milder phenotype. Hum Mutat. 2020 Feb;41(2):420-431. doi: 10.1002/humu.23932) they associate interruptions with late onset of the disease and an atypical phenotype, but not necessarily with a milder phenotype. It would be nice to mention it at this point.
  4. The difficulty in determining the exact size of large expansions, both in DM1 and DM2, by means of TP-PCR is well reflected in the work. Although Southern Blot can overcome this difficulty, it is a time-consuming technique that requires large amounts of DNA, which makes it limiting for its application in routine diagnosis. Regarding this, I think it is worth mentioning that the development of new technologies, such as Long Read Sequencing, will allow, in the near future, characterizing this type of expansions in a deeper and a more precise way.

Author Response

Reviewer 1:

Comments and Suggestions for Authors

In my opinion, it is very appropriate to propose a review work on DM1 and DM2 at this time when different therapeutic strategies, that seem to be quite promising, are emerging and being developed. The work reflects very well the clinical-genetic knowledge that we currently have about these diseases. In addition, the authors appropriately correlate knowledge about the pathophysiological mechanisms of the disease with the concepts underlying therapies in development. In summary, I think this is a good review of DM1 and DM2 that very well collects the most outstanding aspects that are known about these diseases.  However, I would like to suggest some aspects that, in my opinion, will improve the work’s understanding.

Dear Reviewer 1,

Thank you very much for reviewing the manuscript and your comments and suggestions. Please see below my notes/responses under each item.

Best regards

  1. Lines 56-58. “Anticipation is also noted only in DM1, leading to severely affected congenital DM1 children born from mothers who may have a mild form of classic DM1”. A short sentence would be included detailing that the anticipation phenomena appears because CTG repeats greater than 34 are unstable and may expand during meiosis at risk of increasing over successive generations.

I added a sentence and a reference on the role of meiosis.

  1. Line 135: “In DM1, there is a correlation between the number of CTG repeats and the severity of the disease”. Although it is well explained that congenital DM is the most severe form of DM1, I would add in the mentioned sentence that there is a correlation between the number of CTG repeats and the severity and age of onset of the disease.

Phrase “and age of onset” was added in the sentence. This was mentioned both the “clinical features” section as well as in the same section, a few lines after; however, is worth mentioning again.

  1. Line 148: “……interruptions within the CTG repeat array leading to repeat stabilization and potentially milder phenotypes”. It seems that there could be some discrepancies regarding this point. In the following article (Ballester-Lopez A et al. A DM1 family with interruptions associated with atypical symptoms and late onset but not with a milder phenotype. Hum Mutat. 2020 Feb;41(2):420-431. doi: 10.1002/humu.23932) they associate interruptions with late onset of the disease and an atypical phenotype, but not necessarily with a milder phenotype. It would be nice to mention it at this point.

The suggested work was added to the manuscript since they have presented a nice review of “interrupted families” in table 2, some atypical features in the three sisters are interesting and there needs more data on this topic; however, I do not consider this paper a challenger to what has been described about the mitigation effect of interruptions. As the authors point out, the age at onset is very hard to study in DM1 patients and it may take many years for DM1 patients to seek help. The reported sisters presented with a late onset of disease, which can be considered a mild phenotypic feature and no congenital or childhood DM1 was noted in any of their families. The predominant late axial weakness and lack of typical DM1 craniofacial features are very interesting clinically.

  1. The difficulty in determining the exact size of large expansions, both in DM1 and DM2, by means of TP-PCR is well reflected in the work. Although Southern Blot can overcome this difficulty, it is a time-consuming technique that requires large amounts of DNA, which makes it limiting for its application in routine diagnosis. Regarding this, I think it is worth mentioning that the development of new technologies, such as Long Read Sequencing, will allow, in the near future, characterizing this type of expansions in a deeper and a more precise way.

Thank you for pointing this out. Your comment about long-read sequencing technologies and relevant references were added before Table 2 in the “Genetics of Myotonic Dystrophies” section.

Submission Date

17 January 2022

Date of this review

27 Jan 2022 13:46:03

Reviewer 2 Report

The article is clearly written throughout, and readers will find it a useful introduction to DM1 and DM2. However, there are some omissions and some minor errors, especially in the genetics sections, which should be addressed to improve your article throughout before publication.

In general, there are not nearly enough references. For example, in Section 2 (starting on line 44) there are several statements of fact, which are not backed up by references. I would say references are needed for the following at least (paraphrased from your text):

  1. a) Age at disease onset in DM1 is determined by CTG repeat number
  2. b) no such relationship is seen for CCTG repeats in DM2
  3. c) anticipation is observed in DM1 (but not in DM2)

Please could you read through the whole article and check that every time you make a statement of fact along these lines, there is a reference for it? It could be a very general reference, e.g. Peter Harper’s book describing the clinical aspects of DM1, but the article would be a much more useful tool for the non-expert reader if the references were given.

In lines 56-57, please could you amend to say congenitally affected DM1 patients are usually born to affected mothers? In your later section about congenital DM1 you discuss paternal transmission of congenital DM1, but here you imply it’s always maternal, which is obviously not the case.

Table 1: I’m not sure you can categorically state any DM1 symptom is always present (early cataract, myotonia). Some of the variant repeat patients can have very unusual symptoms, so I think it’s too strong a statement. Also, in a hypothetical patient with the late onset form of the disease, if they got cataracts at 65 when their muscular symptoms began, one couldn’t be 100% certain it was DM1-related, rather than just age. It would be better to use “almost always” even for the commonest symptoms.

Sections 2.1, 2.2, 2.3 again need some references

Section 3 Genetics of Myotonic Dystrophies

Lines 135-136 (CTG repeat v. age at disease onset) again needs a reference, as does your quoted repeat length for CDM1.

Line 150 variant repeats NOT variable repeats (please change this-presumably a typo but unhelpful for the reader).

I think you could also include more about variant repeats in DM1. To be fair, the exact role of variant repeats in DM1 pathophysiology has not been determined (lines 150-151), but it would be untrue to say that it means nothing is known. There are now several studies showing that disease onset is delayed when there are variant repeats (Overend 2019, Hum Mol Genet, Pešović 2018, Frontiers in Genetics). In addition, there is evidence of reduced somatic instability in individuals with variant repeats (same three studies). Somatic instability is important for both disease onset and progression. Variant repeats may also cause atypical symptoms. Patients with variant repeats also scored differently (and were less affected) on various outcome measures, including quality of life, muscle and psychometric measures, and changes in brain structure (Miller 2020, Neurol Genet). There is also evidence that variant repeats alter methylation patterns around repeat expansions (and the paper you refer to later, and an earlier study (Santoro, 2015, Biochim Biophys Acta). So, despite how variant repeats change, all the details of the pathology of the disease are not fully understood, differences in somatic instability are clearly important and worth mentioning, even if only briefly. Note I have included papers that come to mind immediately, but I am sure there are others, So please investigate. So while all the details of how variant duplication alters disease pathology are not fully understood, differences in somatic instability are clearly important and worth mentioning, even if only briefly. Note that I have included immediate paper that comes to mind, but I'm sure there are others, so check it out.

Glad you mentioned the importance of two-way bidirectional for testing, as variant repeats is a key point.

The paragraph about repeat instability in DM1 could also be expanded and clarified a bit.

The method used to estimate repeat length is important. If you consider the average repeat length (usually the mode, the middle of the length distribution), it will vary depending on how many repeats the patient inherited, what tissue you examine, and how old they were when you collected the sample. This is why it is useful to try to estimate the inherited repeat length using small-pool PCR. The inherited repeat length is very useful as a predictor of age at disease onset (Morales 2012, Hum Mol Genet), in fact it is the single best predictor (it accounts for about 65-70% of the variability in onset) and is further modified by somatic instability. So, if two people inherit, say, 100 CTGs, and one has repeats that are more unstable than the other, their symptoms will probably start earlier and progress more quickly.

You state that using lymphocytes to measure CTG repeats is a limitation of many studies (line 166-169). I disagree. It’s true that the repeats are relatively stable in lymphocytes, but that fact also implies that in most cases, it is possible to make an estimate of the inherited number of repeats. As I’ve already mentioned, the inherited repeat length is highly predictive for age at onset. The same would not be true of muscle biopsies, for example-even if such tissues were available, they wouldn’t give us more information than lymphocytes, just different information. It’s also important to use tissues that are easily accessible (blood, saliva) and extract the best information we can from what we can get.

 It would also be useful to say something about the role of mismatch repair genes here. Later you say MMR is more likely than DNA replication to be a factor in repeat instability (true) but there is quite a bit of evidence for the involvement of MMR. There are several transgenic mouse studies in both DM1 and HD (very similar in mechanism) where crosses between MMR gene knockouts and DM1 or HD transgenics lose somatic instability and also disease symptoms. Also, polymorphisms in MSH3 have been shown to modify somatic instability (Morales 2016, DNA Repair) and disease severity. So, modifying somatic instability, particularly via MSH3, is an important research field likely to lead to future therapies, and it’s worth including a paragraph or two. Again, I’ve included a few references that immediately came to mind, but there will almost certainly be others.

When you discuss CDM1, (line 170 to 201) it’s also worth mentioning reduced fertility in men with DM1, and the possibility that sperm from affected fathers that have very large expansions (likely, given the range of repeat lengths seen in sperm) might, if they fertilize the egg, might produce embryos that are not viable and result in early miscarriage. These effects could also increase the incidence of CDM1 babies born to affected mothers.

Section 4.2 RAN translation

This needs a major rewrite. For a start, the three reading frames are NOT sense, antisense and RAN. Sense and antisense are transcripts made from the two strands of the DNA-both exist in both DM1 and DM2. Reading frames refer to the register or start position of the codons. For the DM1 repeat, there are three possible reading frames:

CUG|CUG|CUG|CUG|CUG translated would give polyleucine

C|UGC|UGC|UGC|UGC translated would give polycysteine

CU|GCU|GCU|GCU|GCU translated would give polyalanine

An antisense transcript would contain CAG repeats instead, which similarly could be translated into three different peptides (polyglutamine if CAG, polyserine if AGC, polyalanine if GCA). Thus, each transcript could give rise to three peptides. In lines 314-315 you mention polyglutamine aggregates, but this is one of the six possible peptides generated by the existence of multiple reading frames. Your sentence (lines 314 to 315) implies there are multiple reading frames all of which produce polyglutamine, which they don’t! Similarly, for DM2, you mention two of the six possible products poly-(LPAC) and poly-(QAGR) accumulate in brains and are known to be toxic. One is from the sense transcript, the other from an antisense transcript, and they are tetrapeptides because it’s a four base repeat, so it takes 12 bases to get back in frame.

On line 371 you mention ADAR1 but don't explain the acronym - please define it as you have defined the other proteins you listed in the same section.

The summaries of ongoing trials and the preclinical studies supporting them are very well done, informative and clear (section 5 to the end). However, it's still worth checking this part of your review to make sure you've cited references where necessary.

Author Response

Reviewer 2:

Comments and Suggestions for Authors

The article is clearly written throughout, and readers will find it a useful introduction to DM1 and DM2. However, there are some omissions and some minor errors, especially in the genetics sections, which should be addressed to improve your article throughout before publication.

Dear Reviewer 2,

Thank you very much for reviewing the manuscript and your detailed comments and suggestions. Please see below my notes/responses under each item.

Best regards

In general, there are not nearly enough references. For example, in Section 2 (starting on line 44) there are several statements of fact, which are not backed up by references. I would say references are needed for the following at least (paraphrased from your text):

  1. Age at disease onset in DM1 is determined by CTG repeat number

Line 53: “In DM1, higher numbers of CTG repeats are usually associated with an earlier onset of disease and more severe phenotype” I added references regarding this, including one from Dr Harper’s group.

  1. no such relationship is seen for CCTG repeats in DM2

References added.

  1. c) anticipation is observed in DM1 (but not in DM2)

References added.

Please could you read through the whole article and check that every time you make a statement of fact along these lines, there is a reference for it? It could be a very general reference, e.g. Peter Harper’s book describing the clinical aspects of DM1, but the article would be a much more useful tool for the non-expert reader if the references were given.

Additional references added

In lines 56-57, please could you amend to say congenitally affected DM1 patients are usually born to affected mothers? In your later section about congenital DM1 you discuss paternal transmission of congenital DM1, but here you imply it’s always maternal, which is obviously not the case.

Added more clarification

Table 1: I’m not sure you can categorically state any DM1 symptom is always present (early cataract, myotonia). Some of the variant repeat patients can have very unusual symptoms, so I think it’s too strong a statement. Also, in a hypothetical patient with the late onset form of the disease, if they got cataracts at 65 when their muscular symptoms began, one couldn’t be 100% certain it was DM1-related, rather than just age. It would be better to use “almost always” even for the commonest symptoms.

Agree. Corrections made.

Sections 2.1, 2.2, 2.3 again need some references

Additional references added

Section 3 Genetics of Myotonic Dystrophies

Lines 135-136 (CTG repeat v. age at disease onset) again needs a reference, as does your quoted repeat length for CDM1.

Additional references added

Line 150 variant repeats NOT variable repeats (please change this-presumably a typo but unhelpful for the reader).

The typo was corrected

I think you could also include more about variant repeats in DM1. To be fair, the exact role of variant repeats in DM1 pathophysiology has not been determined (lines 150-151), but it would be untrue to say that it means nothing is known. There are now several studies showing that disease onset is delayed when there are variant repeats (Overend 2019, Hum Mol Genet, Pešović 2018, Frontiers in Genetics). In addition, there is evidence of reduced somatic instability in individuals with variant repeats (same three studies). Somatic instability is important for both disease onset and progression. Variant repeats may also cause atypical symptoms. Patients with variant repeats also scored differently (and were less affected) on various outcome measures, including quality of life, muscle and psychometric measures, and changes in brain structure (Miller 2020, Neurol Genet). There is also evidence that variant repeats alter methylation patterns around repeat expansions (and the paper you refer to later, and an earlier study (Santoro, 2015, Biochim Biophys Acta). So, despite how variant repeats change, all the details of the pathology of the disease are not fully understood, differences in somatic instability are clearly important and worth mentioning, even if only briefly. Note I have included papers that come to mind immediately, but I am sure there are others, So please investigate. So while all the details of how variant duplication alters disease pathology are not fully understood, differences in somatic instability are clearly important and worth mentioning, even if only briefly. Note that I have included immediate paper that comes to mind, but I'm sure there are others, so check it out.

This section was expanded and additional references added.

Glad you mentioned the importance of two-way bidirectional for testing, as variant repeats is a key point.

The paragraph about repeat instability in DM1 could also be expanded and clarified a bit.

The method used to estimate repeat length is important. If you consider the average repeat length (usually the mode, the middle of the length distribution), it will vary depending on how many repeats the patient inherited, what tissue you examine, and how old they were when you collected the sample. This is why it is useful to try to estimate the inherited repeat length using small-pool PCR. The inherited repeat length is very useful as a predictor of age at disease onset (Morales 2012, Hum Mol Genet), in fact it is the single best predictor (it accounts for about 65-70% of the variability in onset) and is further modified by somatic instability. So, if two people inherit, say, 100 CTGs, and one has repeats that are more unstable than the other, their symptoms will probably start earlier and progress more quickly.

Note about estimated progenitor allele length (ePAL) and related references added

You state that using lymphocytes to measure CTG repeats is a limitation of many studies (line 166-169). I disagree. It’s true that the repeats are relatively stable in lymphocytes, but that fact also implies that in most cases, it is possible to make an estimate of the inherited number of repeats. As I’ve already mentioned, the inherited repeat length is highly predictive for age at onset. The same would not be true of muscle biopsies, for example-even if such tissues were available, they wouldn’t give us more information than lymphocytes, just different information. It’s also important to use tissues that are easily accessible (blood, saliva) and extract the best information we can from what we can get.

Agree that PAL derived from peripheral blood is more helpful but still such peripheral measurements lack tissue specificity and even 65-70% is not ideal. The phrase meant to show more the complexity of this issue. I removed “One limitation of many DM1 studies is that” for more clarity.

 It would also be useful to say something about the role of mismatch repair genes here. Later you say MMR is more likely than DNA replication to be a factor in repeat instability (true) but there is quite a bit of evidence for the involvement of MMR. There are several transgenic mouse studies in both DM1 and HD (very similar in mechanism) where crosses between MMR gene knockouts and DM1 or HD transgenics lose somatic instability and also disease symptoms. Also, polymorphisms in MSH3 have been shown to modify somatic instability (Morales 2016, DNA Repair) and disease severity. So, modifying somatic instability, particularly via MSH3, is an important research field likely to lead to future therapies, and it’s worth including a paragraph or two. Again, I’ve included a few references that immediately came to mind, but there will almost certainly be others.

Brief discussion about MMR added.

When you discuss CDM1, (line 170 to 201) it’s also worth mentioning reduced fertility in men with DM1, and the possibility that sperm from affected fathers that have very large expansions (likely, given the range of repeat lengths seen in sperm) might, if they fertilize the egg, might produce embryos that are not viable and result in early miscarriage. These effects could also increase the incidence of CDM1 babies born to affected mothers.

For better clarification, a general/introductory comment about the possible role of sperm health was added.

Section 4.2 RAN translation

This needs a major rewrite. For a start, the three reading frames are NOT sense, antisense and RAN. Sense and antisense are transcripts made from the two strands of the DNA-both exist in both DM1 and DM2. Reading frames refer to the register or start position of the codons. For the DM1 repeat, there are three possible reading frames:

CUG|CUG|CUG|CUG|CUG translated would give polyleucine

C|UGC|UGC|UGC|UGC translated would give polycysteine

CU|GCU|GCU|GCU|GCU translated would give polyalanine

An antisense transcript would contain CAG repeats instead, which similarly could be translated into three different peptides (polyglutamine if CAG, polyserine if AGC, polyalanine if GCA). Thus, each transcript could give rise to three peptides. In lines 314-315 you mention polyglutamine aggregates, but this is one of the six possible peptides generated by the existence of multiple reading frames. Your sentence (lines 314 to 315) implies there are multiple reading frames all of which produce polyglutamine, which they don’t! Similarly, for DM2, you mention two of the six possible products poly-(LPAC) and poly-(QAGR) accumulate in brains and are known to be toxic. One is from the sense transcript, the other from an antisense transcript, and they are tetrapeptides because it’s a four base repeat, so it takes 12 bases to get back in frame.

Thank you for your detailed comment. I edited the paragraph. The polyglutamine was (has been) the product studied. I clarified that this is one of the byproducts. There is paucity of data on this topic in DMs.

On line 371 you mention ADAR1 but don't explain the acronym - please define it as you have defined the other proteins you listed in the same section.

“adenosine deaminase acting on RNA 1” was added

The summaries of ongoing trials and the preclinical studies supporting them are very well done, informative and clear (section 5 to the end). However, it's still worth checking this part of your review to make sure you've cited references where necessary.

I added a couple of references.

Submission Date

17 January 2022

Date of this review

28 Jan 2022 18:31:47